# Research on the Impact of Data Density on Memristor Crossbar Architectures in Neuromorphic Pattern Recognition

**DOI:** 10.3390/mi14111990

**Published:** 2023-10-27

**Authors:** Minh Le, Son Ngoc Truong

**Affiliations:** Faculty of Electrical and Electronics Engineering, Ho Chi Minh City University of Technology and Education, Ho Chi Minh City 70000, Vietnam; leminh@hcmute.edu.vn

**Keywords:** pattern recognition, memristor crossbar circuit, neuromorphic computing

## Abstract

Binary memristor crossbars have great potential for use in brain-inspired neuromorphic computing. The complementary crossbar array has been proposed to perform the Exclusive-NOR function for neuromorphic pattern recognition. The single crossbar obtained by shortening the Exclusive-NOR function has more advantages in terms of power consumption, area occupancy, and fault tolerance. In this paper, we present the impact of data density on the single memristor crossbar architecture for neuromorphic image recognition. The impact of data density on the single memristor architecture is mathematically derived from the reduced formula of the Exclusive-NOR function, and then verified via circuit simulation. The complementary and single crossbar architectures are tested by using ten 32 × 32 images with different data densities of 0.25, 0.5, and 0.75. The simulation results showed that the data density of images has a negative effect on the single memristor crossbar architecture while not affecting the complementary memristor crossbar architecture. The maximum output column current produced by the single memristor crossbar array decreases as data density decreases while the complementary memristor crossbar array architecture provides stable maximum output column currents. When recognizing images with data density as low as 0.25, the maximum output column currents of the single memristor crossbar architecture is reduced four-fold compared with the maximum currents from the complementary memristor crossbar architecture. This reduction causes the Winner-take-all circuit to work incorrectly and will reduce the recognition rate of the single memristor crossbar architecture. These simulation results show that the single memristor crossbar architecture has more advantages compared with the complementary crossbar architecture when the images do have not many different densities, and none of the images have very low densities. This work also indicates that the single crossbar architecture must be improved by adding a constant term to deal with images that have low data densities. These are valuable case studies for archiving the advantages of single memristor crossbar architecture in neuromorphic computing applications.

## 1. Introduction

Memristor was mathematically proposed in 1971 by Prof. L. O. Chua as basic as the three circuit elements, namely the resistor, inductor, and capacitor [1]. The first practical memristor device was introduced by R. S. William and several colleagues at Hewlett-Packard Laboratories in 2008 [2]. The conductance of a memristor, also known as memristance, can be modified by programming pulses and has the ability to be maintained, making the memristor an ideal device for modelling the synaptic plasticity of biological neuronal systems [3,4]. Furthermore, with 2D array and 3D array structures [5,6,7,8,9], memristor crossbar arrays have become an emerging technology for high-density neuromorphic computing systems, as an alternative to CMOS technology that is unquestionably approaching the physical scaling limits [10,11].

Hardware implementations of neural computing using memristor crossbars have achieved much success in the last decade [12,13,14,15,16,17]. Since the multiplication and accumulation operations can be performed using Kirchoff’s law and Ohm’s law at the circuit level, the results can be obtained in a single step, leading to a significant improvement in computational speed, energy consumption, and area occupancy [18]. Although the memristor crossbar has many advantages, the implementation of neural computing using memristor crossbar faces many challenges, caused by non-ideal device parameters, for example, programming variation, state-stuck devices, conductance drift, and device variability [19,20]. The binary memristor crossbar, in which memristor has only two states: low resistance and high resistance states, becomes more feasible for neuromorphic computing [13,16,21,22]. The binary memristor crossbar can perform the cognitive task of pattern recognition, which is the process that matches information from a stimulus with information retrieved from the memory [23]. Several brain-inspired neuromorphic computing circuits employing binary memristor crossbar arrays for neuromorphic pattern recognitions such as speech recognition [24,25] and image recognition [26,27] have been recently proposed. With the high ratio of high resistance state to low resistance state, binary memristor arrays are more efficient for implementing brain-inspired neuromorphic computing for pattern recognition applications in terms of power consumption, and noise and device variation tolerance, compared with analog memristor crossbar arrays.

The first interesting architecture of binary memristor crossbar for brain-inspired neuromorphic computing is the complementary crossbar that performs the logical function of Exclusive-NOR for speech and image recognition [24]. The twin crossbar architecture is a modified version of the complementary crossbar for low-power neuromorphic image recognition [26]. The single crossbar architecture is then an optimized version of the complementary crossbar and the twin crossbar by shortening the Exclusive-NOR function [27]. The single memristor crossbar architecture has more advantages in terms of area occupancy, power consumption, and fault tolerance. The single memristor crossbar is a potential piece of architecture for neuromorphic image recognition because it can save area occupancy and power consumption, compared to the complementary and twin crossbar architectures.

In previous work, to obtain the single crossbar architecture, a constant term in the expanded function of the Exclusive-NOR is omitted. Because all constant terms of all columns are omitted, so it does not affect the identification of the winner. The single memristor crossbar circuit was tested with 10 binary images. The tested images have a high number of 1 bits. Single crossbar architecture has not been tested with images with the low number of 1 bits. Bit 1 of binary image data is represented by a low resistance state memristor, which mainly produces the output column current in single memristor crossbar architecture. If the number of 1 bits is low, the output currents are all very small, which can impact the accuracy of output decision circuit. In this work, we find out the impact of data density on the operation of the single memristor crossbar architecture, in which a constant term of the expanded function of the Exclusive-NOR is omitted. This research shows an interesting result that the single memristor crossbar architecture has the advantage for images with high density, but does not work well with low-density images.

## 2. The Complementary Memristor Crossbar Architecture and the Single Memristor Crossbar Architecture for Neuromorphic Pattern Recognition

A complementary memristor crossbar architecture has been proposed for the cognitive task of pattern recognition based on the Exclusive NOR operation to measure the similarity between the input pattern and the stored patterns. Complementary memristor crossbar architecture is composed of two complementary crossbar arrays, as conceptually shown in Figure 1. The column outputs are obtained by the Exclusive NOR operation between the input vector and the column vectors [24]:(1)$$Y=\bar{A⊕M}=AM+A^{\prime }M^{\prime }=A\cdot \left(M+\right)+A^{\prime }\cdot \left(M-\right)$$

In Equation (1), A is the input vector, M+ and M− represent the memristor crossbar and its inversion, which consists of inverted elements of M+, respectively. The block diagram and schematic of the complementary memristor crossbar architecture are shown in Figure 1.

Figure 1a conceptually shows a block diagram of the complementary crossbar architecture for recognizing m patterns. In Figure 1a, the input vector A has the size of 1 × *n*, the M+ and M− are the two complementary memristor arrays with the size of *n* × *m* in which m patterns are pre-stored for later recognition. Each pattern is saved in one column of the arrays, in the format of binary data. A memristor in one column of the M+ array may be set at either a high resistance state (HRS) or a low resistance state (LRS) when storing a bit 0 or a bit 1, respectively. The M− array contains memristors that have inverted values with corresponding memristors in M+. For example, if the $M_{0,0}$ memristor in the M+ array has the value of HRS, the ${M\prime }_{0,0}$ memristor in the M− array will have the value of LRS. The M+ array and M− array can be described in matrices as follows:(2)$$\begin{array}{c} M+=\left[\begin{array}{cccc} M_{0,0} & M_{0,1} & \ldots & M_{0,\left(m-1\right)}\\ M_{1,0} & M_{1,1} & \ldots & M_{1,\left(m-1\right)}\\ \vdots & \vdots & \vdots & \vdots \\ M_{\left(n-1\right),0} & M_{\left(n-1\right),1} & \ldots & M_{\left(n-1\right),\left(m-1\right)} \end{array}\right]\\ M-=\left[\begin{array}{cccc} {M^{\prime }}_{0,0} & {M^{\prime }}_{0,1} & \ldots & {M^{\prime }}_{0,(m-1)}\\ {M^{\prime }}_{1,0} & {M^{\prime }}_{1,1} & \ldots & {M^{\prime }}_{1,(m-1)}\\ \vdots & \vdots & \vdots & \vdots \\ {M^{\prime }}_{\left(n-1\right),0} & {M^{\prime }}_{\left(n-1\right),1} & \ldots & {M^{\prime }}_{\left(n-1\right),(m-1)} \end{array}\right] \end{array}$$

The input vector A is applied to the M+ array, and its inversion vector, A′, is applied to the M− array to implement the Exclusive-NOR function between A and M as discussed in Equation (1) in order to obtain the following results:(3)$$\begin{array}{c} Y=\left[\begin{array}{cccc} a_{0} & a_{1} & \ldots & a_{\left(n-1\right)} \end{array}\right]\cdot \left[\begin{array}{cccc} M_{0,0} & M_{0,1} & \ldots & M_{0,\left(m-1\right)}\\ M_{1,0} & M_{1,1} & \ldots & M_{1,\left(m-1\right)}\\ \vdots & \vdots & \vdots & \vdots \\ M_{\left(n-1\right),0} & M_{\left(n-1\right),1} & \ldots & M_{\left(n-1\right),\left(m-1\right)} \end{array}\right]\\ +\left[\begin{array}{cccc} {a^{\prime }}_{0} & {a^{\prime }}_{1} & \ldots & {a^{\prime }}_{\left(n-1\right)} \end{array}\right]\cdot \left[\begin{array}{cccc} {M^{\prime }}_{0,0} & {M^{\prime }}_{0,1} & \ldots & {M^{\prime }}_{0,(m-1)}\\ {M^{\prime }}_{1,0} & {M^{\prime }}_{1,1} & \ldots & {M^{\prime }}_{1,(m-1)}\\ \vdots & \vdots & \vdots & \vdots \\ {M^{\prime }}_{\left(n-1\right),0} & {M^{\prime }}_{\left(n-1\right),1} & \ldots & {M^{\prime }}_{\left(n-1\right),(m-1)} \end{array}\right]\\ =\left[\begin{array}{cccc} i_{0} & i_{1} & \cdots & i_{m-1} \end{array}\right] \end{array}$$
where $Y=\left[\begin{array}{cccc} i_{0} & i_{1} & \cdots & i_{m-1} \end{array}\right]$ is the output vector that contains m output column currents.

The output current is then fed into a Winner-take-all circuit, which determines the maximum output current. If the Winner-take-all circuit shows that $i_{k}$ is the maximum output current, it means that the input vector A best matches the pattern stored in column kth of the arrays.

Figure 1b represents the schematic of a complementary memristor crossbar circuit for recognizing ten black and white images with the size of 32 × 32. Each image is converted into a vector of size 1024 × 1 and stored in one column of the M+ array while its inverted vector is stored in the corresponding column of the M− array. The input image represented by vector $A=\left[\begin{array}{cccc} a_{0} & a_{1} & \ldots & a_{1023} \end{array}\right]$ and its inversion vector A′, are applied to the M+ array and the M− array as presented in Equation (3). The output column current $i_{k}$ is then copied by a current mirror circuit, and makes the pre-charged capacitor $C_{k}$ discharge. When the capacitor $C_{k}$ discharges, the voltage ${VC}_{k}$ decreases either fast or slowly depending on the value of the current $i_{k}$. If the current $i_{k}$ is large, the capacitor $C_{k}$ discharges fast and the voltage ${VC}_{k}$ decreases fast. Ten discharging voltages, ${VC}_{0}$ to ${VC}_{9}$, are then compared to each other using the Winner-take-all circuit to find the fastest one. The schematic of the Winner-take-all circuit is shown in Figure 2 [24].

In the Winner-take-all circuit, ten comparators receive ten discharging voltages, from ${VC}_{0}$ to ${VC}_{9}$, and compare these voltages with the reference voltage $V_{REF}$. When a voltage ${VC}_{k}$ decreases to below the $V_{REF}$, the output $D_{k}$ changes to high while the other outputs remaining low. This means that if ${VC}_{k}$ is the fastest discharging voltage, the comparators set only $D_{k}$ to high. The Pulse Generator then produces a locking pulse after a delaying time to set the ${Output}_{k}$ to high by the flip-flop ${FF}_{k}$. The ${Output}_{k}$ becomes high, while the other outputs remaining low indicates that the input vector A matches the pattern in the column kth of the memristor arrays.

The single memristor crossbar architecture was proposed by utilizing the Exclusive-NOR function with only one memristor array [27]. The Exclusive-NOR function can be expanded as follows:(4)$$\begin{array}{c} Y=\bar{A⊕M}=AM+A^{\prime }M^{\prime }\\ =AM+A\prime (1-M)\\ =\left(A-A^{\prime }\right)M+A^{\prime } \end{array}$$

In Equation (4), A′ is a constant term for all columns and can be ignored because this term does not affect the determination of the maximum output current. The optimized Exclusive-NOR function for the single memristor crossbar architecture is expressed as:(5)$$\begin{array}{c} Y=\bar{A⊕M}=B\cdot M\\ where\;B=\left(A-A^{\prime }\right) \end{array}$$
or:(6)$$\begin{array}{c} Y=\left[\begin{array}{cccc} b_{0} & b_{1} & \ldots & b_{\left(n-1\right)} \end{array}\right]\cdot \left[\begin{array}{cccc} M_{0,0} & M_{0,1} & \ldots & M_{0,\left(m-1\right)}\\ M_{1,0} & M_{1,1} & \ldots & M_{1,\left(m-1\right)}\\ \vdots & \vdots & \vdots & \vdots \\ M_{\left(n-1\right),0} & M_{\left(n-1\right),1} & \ldots & M_{\left(n-1\right),\left(m-1\right)} \end{array}\right]\\ =\left[\begin{array}{cccc} i_{0} & i_{1} & \cdots & i_{m-1} \end{array}\right] \end{array}$$

In Equation (5), $B=\left[\begin{array}{cccc} b_{0} & b_{1} & \cdots & b_{\left(n-1\right)} \end{array}\right]$ is the bipolar input vector generated from subtraction (A−A′) and contains the values 1 and −1. For example, if the input vector A is $A=[\begin{array}{ccc} 0 & 1 & 0 \end{array}]$, A′ will be $A\prime =[\begin{array}{ccc} 1 & 0 & 1 \end{array}]$ and (A−A′) will result $B=[\begin{array}{ccc} -1 & 1 & -1 \end{array}]$. Therefore, single memristor crossbar architecture employs only one memristor array along with a unipolar-to-bipolar Convertor, as shown in Figure 3.

Figure 3a shows the block diagram of the single memristor crossbar architecture for recognizing m patterns and Figure 3b represents the schematic of the single memristor crossbar architecture for recognizing ten 32 × 32 binary images. The input vector A is first turned into the bipolar input vector B by the Unipolar to bipolar Convertor. The bipolar input vector B is next applied to the single memristor array where ten patterns are stored to obtain the output column currents, the $i_{0}$ to $i_{9}$, as expressed in Equations (5) and (6). The output column currents are finally compared to each other by the Winner-take-all circuit to find the maximum output column current $i_{k}$. Here, the input vector A best matches the pattern pre-stored in kth column of the single memristor array.

So far, we can see that the single memristor crossbar array with bipolar input has the same functionality as the complementary memristor crossbar architecture for pattern recognition based on Exclusive-NOR operation. In Equation (4), A is the input vector, M is the memristor array in which images are stored in columns. We apply the input vector to the array and obtain the output column currents. The winning column is identified as the maximum column current by using a digital Winner-take-all circuit [24]. For a particular input, all columns in Equation (4) are added a term of A′; thus, the existence of A′ does not affect the determination of the maximum column current. Based on this inference, it is possible to omit the constant term of A′ to obtain Equation (5). However, in Equation (5), if the input vector A has a large number of 1 bits (defined as high density), meaning A′ has small number of 1 bits (defined as low density), the column currents are all high. If the input vector A has low density, meaning A′ has high density, omitting A′ leads to all column currents are very low. In the CMOS circuit, it is difficult to determine the maximum current when all currents are very low or all currents are very high because CMOS transistors have threshold and saturation voltages. Therefore, the single memristor crossbar architecture becomes a problem when the input images have fewer 1 bits.

## 3. Simulation and Results

The circuit simulations were performed to test the impact of data density on the performance of single memristor crossbar and the complementary memristor crossbar architectures. The simulations were performed using the SPECTRE circuit simulation provided by Cadence Design Systems Inc, San Jose, CA, USA [28]. Memristors were modeled using Verilog-A [29,30]. Memristor model and parameters are chosen to fit the practical memristor device presented in Figure 4 [29,30]. Figure 4 shows a hysteresis behavior of a real memristor based on the film structure of Pt/LaAlO_3_/Nb-doped SrTiO_3_ stacked layer and a memristor model that can be used to describe various memristive behaviors [29,30].

As discussed in the previous section, it is essential to analyze the impact of data density of patterns on the complementary and the single memristor crossbar architectures. The data density of a binary image is defined as the percentage of bit 1 s in the image data. In particular, images with high data density will have a higher number of 1 bits, whereas images with low data density will have fewer 1 bits. In this paper, ten images are used to analyze the impact of data density on the performance of memristor crossbar architectures. The original images are presented in Figure 5.

The original images are grayscale images with the size of 32 × 32. Binary images are produced by thresholding grayscale images. By varying the threshold, we obtain images with different data densities. The first three images (#0, #1, and #2) have a low data density of 0.25, the next three images (#3, #4, #5) have a moderate data density of 0.5, and the last four images (#6, #7, #8, and #9) have a high density of 0.75. These different data density images are then vectorized to the size of 1024 × 1 and stored in the memristor arrays of the complementary crossbar architecture and the single crossbar architecture. Each image is stored in a column of the array. Binary images with different data density produced by thresholding grayscale images are shown in Figure 6.

In Figure 6a, a low data density of 0.25 means that the number of bits 1 accounts for 25% of the total number of pixels in the image. In Figure 6b, the images have equal numbers of white pixels and black pixels, and images in Figure 6c have a greater number of white pixels than black pixels.

Binary images are represented by vectors of binary values. Each image is stored in one column of the memristor array for single crossbar architecture. For complementary crossbar architecture, each image is stored in two columns, one column in the memristor array and the other in the inverted memristor array, as mentioned before. Binary value 0 is represented by the high resistance state (HRS) memristor and binary 1 is represented by the low resistance state (LRS) memristor in the crossbar array. The HRS and LRS are 1 MΩ and 10 KΩ, respectively. The binary values 0 and 1 in the input vector are mapped to input voltages of 0 V and 1 V, respectively. The input image represented by the vector of input voltage is applied to the crossbar circuit. The output currents are produced at the bottom of columns according to the Ohm’s law and the Kirchoff’s current law. These output column currents are then compared to each other using a Winner-take-all circuit to determine the maximum column current, corresponding to the column containing the pre-stored image that best matches the input image. The Winner-take-all circuit is based on the discharge speeds of pre-charged capacitors, which are controlled by the output column currents, to find the fastest discharging capacitor. Therefore, the values of output column currents play an important role in the recognition accuracy of the memristor crossbar array architectures. The output column currents when recognizing ten input images with different data densities are shown in Figure 7.

Figure 7a reveals that the complementary crossbar architecture produces the same amount of maximum column currents when recognizing 10 images (from #0 to #9) which have different data densities. In other words, the maximum output column current of the complementary crossbar architecture does not depend on the data density of the input images and the stored images. In particular, although the data densities of input images are varied from 0.25 to 0.75, the maximum output column currents are stable at above 100 mA. The reason for these stable maximum output column currents is that the complementary crossbar architecture employs two complementary memristor arrays: the M+ memristor array and the M− memristor array which contains memristors with inverted values of the corresponding memristors in the M+ array. When a low data density image is stored in the M+ memristor array, its inverted image or the complementary high data density image would also be stored in the M− memristor array and vice versa. An output column current is the sum of corresponding output currents from the M+ and M− arrays; therefore, the maximum output column current remain unchanged regardless of the input images with different data densities.

In contrast, with the single memristor crossbar architecture, the output column currents reduce when the data densities of input images are decreased, as shown in Figure 7b. In particular, when the data density of input images is as low as 0.25 (images #0, #1, #2), the maximum output column currents decreased as much as four times in comparison with the complementary crossbar architecture and the other column currents are 0. The reason for this result is described by Equation (5). In Equation (5), the parameter A′ is omitted because it is a constant. Although this dismissing is mathematically true for implementing the Exclusive-NOR function with the single memristor crossbar array, it causes a reduction by an amount of A′ at every output column current. In addition, the subtraction in Equation (5) can yield negative values when the input image has few white pixels or low data density, and these negative values do not generate any current to output column currents. Therefore, when recognizing input images with a low data density of 0.25 by the single crossbar architecture, the maximum column current reduces about 4 times in comparison with by the complementary crossbar architecture, and the rest column currents are 0. When the data density is 0.5 (images #3, #4, #5) and 0.75 (images #6, #7, #8, #9), the maximum output column currents produced by the single crossbar architecture are also decreased, equal to around 0.5 and 0.75; the largest one is generated by the complementary crossbar architecture.

Because the Winner-take-all circuit is based on the output column currents, this reduction in the maximum output column current of the single memristor crossbar architecture should be considered. As represented in previous section, the output column currents from memristor crossbars cause the pre-charged capacitors, the $C_{0}$ to $C_{9}$, to discharge at different speeds. When an output column current is high, it makes the corresponding capacitor discharge fast, and a capacitor will discharge slowly when the corresponding column current is low. The discharging voltages, the ${VC}_{0}$ to ${VC}_{9}$, from the pre-charged capacitors is fed into the winner-take-all circuit to determine the maximum output current, corresponding to fastest discharging voltage. When the fastest discharging voltage degrades to below the reference voltage of 0.5 V, it makes the Pulse Generator create a pulse to lock the wining output among all outputs, from ${Output}_{0}$ to ${Output}_{9}$. If the ${Output}_{k}$ becomes 1 while the others are 0, it indicates that ${VC}_{k}$ is the fastest discharging voltage or the input image best matches the pattern pre-stored in the kth column.

We next analyze the discharging voltages, from ${VC}_{0}$ to ${VC}_{9}$, which are produced by the single memristor crossbar array corresponding to different data density input images. The discharging voltages when recognizing the image #6 (with data density of 0.75) and the image #0 (with data density of 0.25) with the single crossbar architecture are shown in Figure 8.

As shown in Figure 8a, when recognizing image #6, which has high data density of 0.75, using the single memristor crossbar array, the pre-charged capacitor $C_{6}$ discharges fastest. The discharging voltage ${VC}_{6}$ decreases fastest to 0.5 V after around 0.3 ns while the others discharging voltages keep as high as above 0.7 V. In the Winner-take-all circuit, the reference voltage of comparators is set at $V_{REF}=0.5V$. Therefore, the comparator $i_{6}$, which received the ${VC}_{6}$ voltage, would set the output $D_{6}$ to high and the Pulse Generator could finally create a locking pulse to lock the ${Output}_{6}=1$ to indicate that the output column current $i_{6}$ is the maximum.

In Figure 8b, when recognizing image #0 (data density is 0.25) using the single memristor crossbar array, after the same period time of 0.3 ns, there is no discharging voltage which decreases below the reference voltage of $V_{REF}=0.5V$. This means that the Pulse Generator could not create a locking pulse and the Winner-take-all circuit could not determine which column current is the maximum after the same period of time as when recognizing image #6 with a high data density of 0.75. These results prove that low data density input images can cause the single memristor crossbar architecture to recognize incorrectly and, therefore, degrade the recognition rate.

## 4. Discussion

The single memristor crossbar is an optimized crossbar architecture for brain-inspired neuromorphic computing. The single memristor crossbar architecture consumes less power and occupies smaller area than the complementary memristor crossbar architecture. In addition, using only one memristor crossbar array can improve the fault tolerance of the memristor crossbar circuit. Here, the cross-point fault is one of the main causes that significantly reduces the accuracy of the memristor crossbar-base neuromorphic circuits [27]. In this paper, we figured out that the single crossbar works well if images have larger number of 1 bits. By contrast, if images have fewer 1 bits, the complementary crossbar architecture performs the image recognition better than the single crossbar architecture. When the input images have data density as low as 0.25, the maximum output column currents obtained by the single memristor crossbar architecture reduce about four times in comparison with the complementary crossbar architecture. The Winner-take-all circuit could not determine the maximum current, leading to the degradation of the recognition rate of the single memristor crossbar architecture. The discoveries from this study are twofold: First, the single memristor crossbar is effective in neuromorphic image recognition provided all images must have high data density. Second, to accommodate images with a low data density, the architecture of the single memristor crossbar must be improved to contain the constant term in the expression of Exclusive-NOR function. These are valuable case studies for archiving the advantages of single memristor crossbar architecture in neuromorphic computing applications.

## 5. Conclusions

In this work, we present the impact of data density on the performance of the single memristor crossbar architecture and the complementary memristor crossbar architecture. The impact of data density on the performance of single crossbar architecture is mathematically figured out by analyzing the effect of the omitted constant term in the Exclusive-NOR operation. The observation is then verified by the circuit simulation for the recognition of images with different levels of density. The complementary crossbar architecture consumes more power and occupies a larger area compared with the single crossbar architecture; however, the complementary crossbar architecture does not depend on data density. Otherwise, single crossbar consumes less power and occupies smaller area than complementary crossbar but single crossbar degrades the performance with low data density images. This work recommends that to ensure the single crossbar architecture works correctly for binary image recognition application, binary images must have high number of 1 bits. Finally, this work also indicates that the single crossbar architecture must be improved by adding a constant term to deal with images that have low data densities. These are valuable case studies for archiving the advantages of single memristor crossbar architecture in neuromorphic computing applications.

## Figures and Tables

**Figure 1 micromachines-14-01990-f001:**
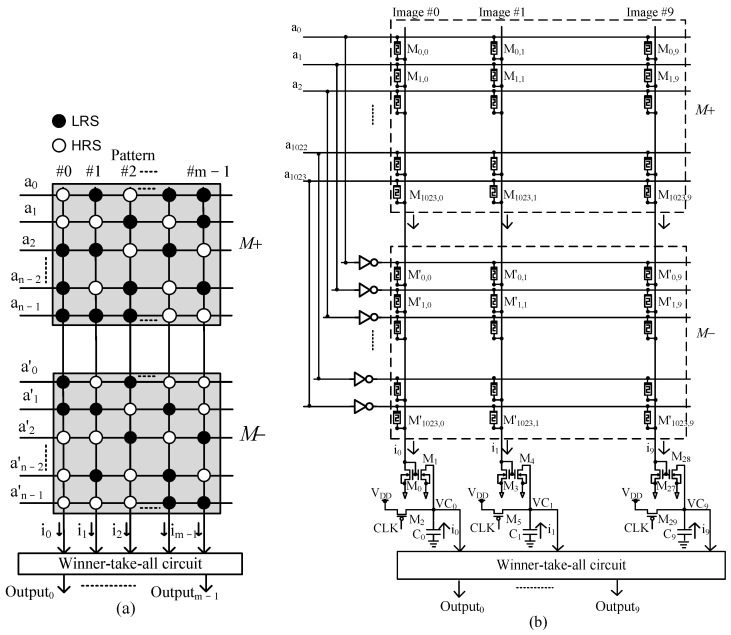
(**a**) The block diagram and (**b**) the schematic of the complementary memristor crossbar architecture for neuromorphic pattern recognition.

**Figure 2 micromachines-14-01990-f002:**
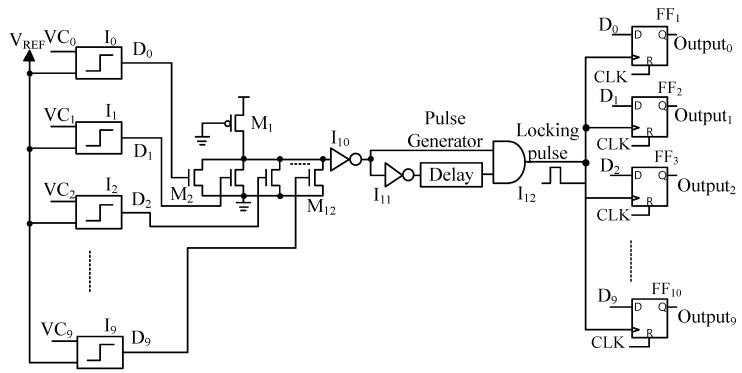
The schematic of the Winner-take-all circuit.

**Figure 3 micromachines-14-01990-f003:**
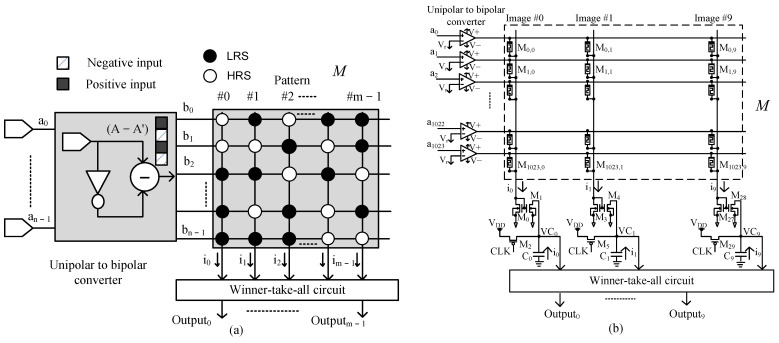
(**a**) The block diagram and (**b**) the schematic of the single memristor crossbar architecture for neuromorphic pattern recognition.

**Figure 4 micromachines-14-01990-f004:**
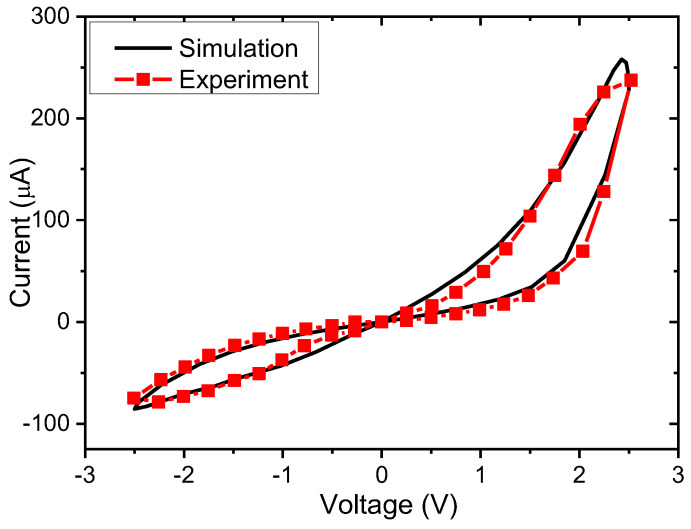
The memristor’s current–voltage characteristic measured from the real device and the memristor’s behavior model [29].

**Figure 5 micromachines-14-01990-f005:**
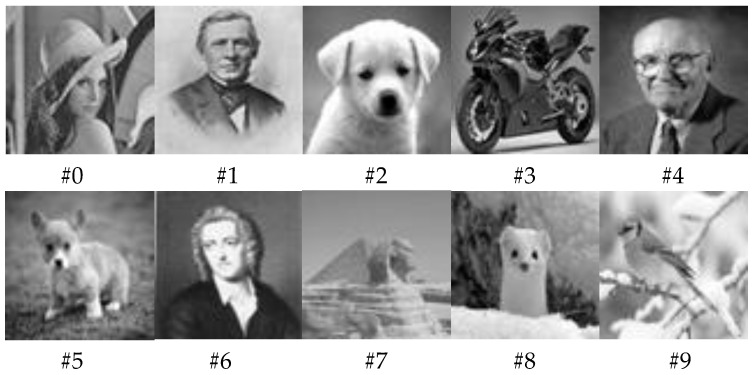
Ten original grayscale images, numbered from image number 0 (#0) to image number 9 (#9).

**Figure 6 micromachines-14-01990-f006:**
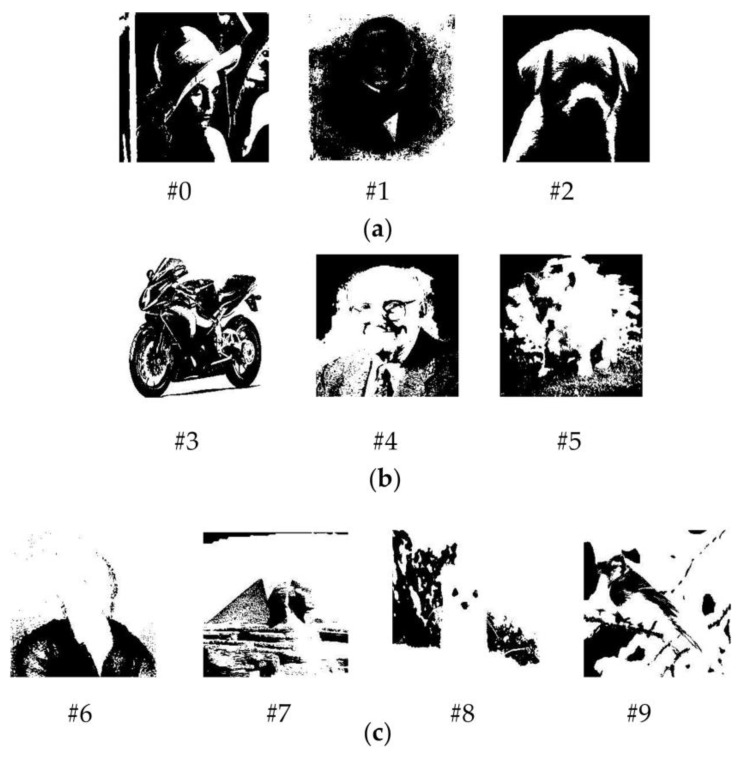
Ten black and white images, numbered from image number 0 (#0) to image number 9 (#9), with different data densities: (**a**) with data density of 0.25, (**b**) with data density of 0.5, (**c**) with data density of 0.75.

**Figure 7 micromachines-14-01990-f007:**
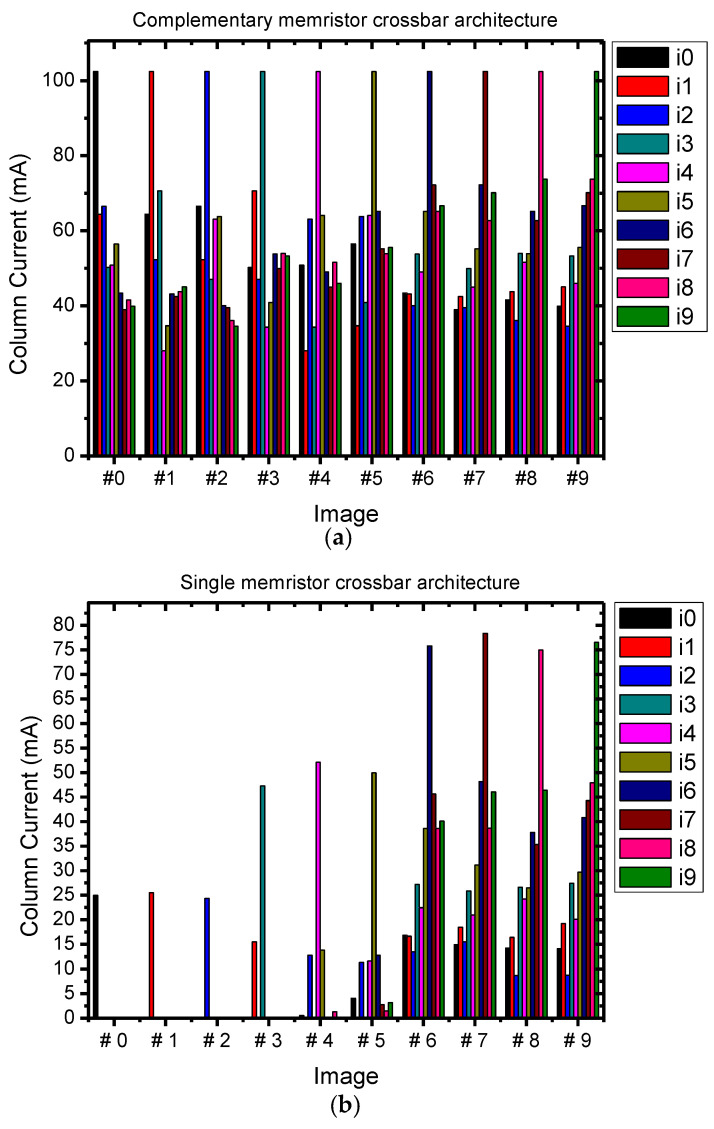
The output column currents when recognizing images from #0 to #9 with: (**a**) the complementary crossbar architecture, (**b**) the single crossbar architecture.

**Figure 8 micromachines-14-01990-f008:**
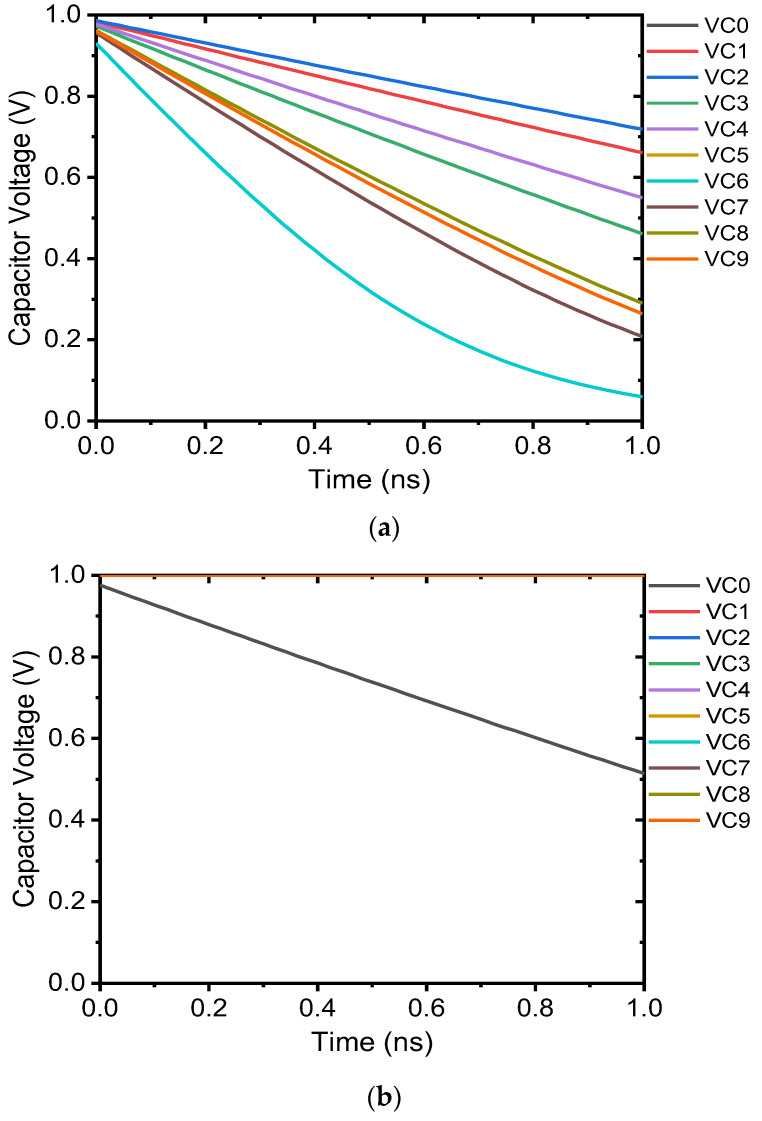
The discharging voltages of pre-charged capacitors when recognizing images with the single memristor crossbar array: (**a**) when recognizing image #6 with high data density of 0.75, (**b**) when recognizing image #0 with low data density of 0.25.

## Data Availability

The data presented in this study are available on request from the corresponding author. The data are not publicly available due to the privacy.

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
