# Peer review of "Research on the Impact of Data Density on Memristor Crossbar Architectures in Neuromorphic Pattern Recognition"

_micromachines, 2023, doi:10.3390/mi14111990_

Round 1

Reviewer 1 Report

Comments and Suggestions for Authors

In this work, the authors provide the sutdy to two different binary memristor crossbar. Bacsiicaly, i think the contents of the work is more or less out of the scope of this special issue but more like some discussion to the cmputation arcthetcture.  While on the techinical ascpect, it provide some information and more inputs for the understanding to the memristor crossbar array. However, these results seems trivial and with relative less novelty. Hence over all, it could be suggsetd to submmited to an more specific jounral other than the high-level comprehensive Jounral like Micromechine

Author Response

Dear Reviewer,

We really appreciate all your questions, comments, and suggestions for us to improve the manuscript.

We have revised our work based on reviewer’s recommendations. Please find the details of our revisions and responses to your comments in the following parts of this letter.

Reviewer 2 Report

Comments and Suggestions for Authors

This article statistically presents the impact of data density on the single memristor crossbar architecture for neuromorphic image recognition and compares it with the complementary memristor crossbar. However, some problems need more specific explanations.

1.      Is the input vector referring to voltage? In that case, how does high voltage correspond to logic 1, and low voltage correspond to logic 0? The diagram lacks corresponding explanations.

2.      According to the description in the article, does this mean that 2^n columns are needed to find a complete match?

3.      How is the binarization of the image achieved? What is the basis for distinguishing data density differences?

4.      If there are multiple output voltages with almost identical attenuation, how can they be distinguished?

5.      Can specific comparisons be provided for power consumption, area, and recognition accuracy? Has chip validation been taped out?

Comments on the Quality of English Language

need to be improved

Author Response

(The authors gave the same response as above.)

Reviewer 3 Report

Comments and Suggestions for Authors

In this manuscript, the authors investigate the impact of data density on the performance of two memristor crossbar architectures: the single memristor crossbar architecture and the complementary memristor crossbar architecture. They find that the single crossbar consumes less power but degrades the performance with low data density images than the complementary crossbar.

The authors' findings are interesting and may be of interest to some readers. However, there is a major concern that needs to be addressed before this work can be published in this journal. 

1. The scientific depth of this study could be significantly improved by the authors providing a more detailed justification of the originality of their work compared to previous reports on complementary memristor crossbar architectures. The authors should explain how this study is different from previous studies.

2. The authors could also be more helpful to readers by providing more explanation about the reasons for utilizing memristors in this architecture. What are the advantages of using memristors over other types of devices? How do they contribute to the overall performance of the system?

3. In addition, the manuscript should also describe details such as the I-V curve, pulse response, reliability, and other characteristics of the employed memristor.

4. More detailed definitions are needed for the parameters "low density data" and "complementary memristor crossbar architecture."

5. In conclusion, the results do not seem to be statistically analyzed.

Author Response

(The authors gave the same response as above.)

Round 2

Reviewer 1 Report

Comments and Suggestions for Authors

After the revision, the draft could be reconsidered, However, some critical concerns shoud be adressed

1) More details or experimatal eviddences to show that the model memristive system emploied for the simulation is truly ionic-switchable; some discusssions and references about the how model the device by verilog-A is helpful for readers

2) Some instances for the neurinic applications of the employied memristive device is valuable to enhance the scope of the manuscript.

Author Response

(The authors gave the same response as above.)

Reviewer 2 Report

Comments and Suggestions for Authors

I suggest the authors can add the discussion part about the imperfection effects of binary memristor on the neural network accuracy as it is essential to the actual applications.

Comments on the Quality of English Language

The language can be polished further to be easily understandable.

Author Response

(The authors gave the same response as above.)

Reviewer 3 Report

Comments and Suggestions for Authors

The authors evaluated the impact of low data density on the performance of two types of memristor crossbar architectures. While the results may be interesting to some readers, I believe they are out of scope for this special issue, "New Advances in Ionic-Drift Resistive Switching Memory and Neuromorphic Applications." The topic seems more closely aligned with circuit simulation or computation architecture. On the technical side, the paper provides some information and insights into memristor crossbar arrays. However, the results seem relatively lacking in novelty. Overall, I suggest the authors submit their paper to a more specific journal. 

Author Response

(The authors gave the same response as above.)

Round 3

Reviewer 1 Report

Comments and Suggestions for Authors

the revised draft can be considered.